# Valuing the Accessibility of Green Spaces in the Housing Market: A Spatial Hedonic Analysis in Shanghai, China

Shenglin Ben [1,2], He Zhu [2], Jiajun Lu [1,*] and Renfeng Wang [2]

1 International Business School, Zhejiang University, Haining 314400, China
2 School of Management, Zhejiang University, Hangzhou 310058, China
* Correspondence: jiajunlu@intl.zju.edu.cn

**Abstract:** As a crucial determinant of urban livability, the importance of access to high-quality green spaces has long been recognized for achieving sustainable urbanization. In urban areas, higher values are usually placed in residential properties with higher accessibility to green spaces. Using housing transaction data from as many as 3388 residential communities in Shanghai coupled with high-resolution satellite data of urban green spaces, we comprehensively examine the relationship between residential property values and the accessibility of both community-owned and public green spaces. We find, in instrumental-variable estimations, that: (1) home prices, on average, increase by 0.17% if the overall green space accessibility rises by 1%; and that (2) a 1% increase in the green ratio within a community raises property values by 0.46%. Moreover, the number of accessible green spaces, area of accessible green spaces, and distance to the nearest green spaces have positive impacts on home values separately. We also find strong spatial dependence in urban green spaces and unobserved price determinants, as well as heterogeneity by location, property value, and housing type. Our empirical findings provide valuable guidance for real estate developers and local governments in valuing environmental amenities and urban planning in the context of a residential housing market.

**Keywords:** urban green spaces (UGS); housing market; spatial hedonic pricing; instrumental variable approach; Shanghai

## 1. Introduction

In the pursuit of sustainable development goals, cities have emerged as critical stakeholders in providing environmental amenities. The global trend of rapid urbanization, exemplified by the case of China, underscores the critical need to strike a delicate balance between economic growth and environmental preservation. Factors such as air quality, water pollution, noise levels, and the availability of green spaces have gained significance in shaping healthier and more pleasant urban environments [1–3]. Among these factors, the significance of green spaces has garnered considerable attention from both academia and industry, owing to their multifaceted benefits [4–6].

The data depicted in Figure 1 reveal a significant rise in residents' incomes, with the per capita wage index increasing from 100 to 228.5 over the past decade at an annualized rate of 9.6%. This substantial increase in income has led to a corresponding expansion of residents' expectations for an enhanced quality of life. Furthermore, Figure 1 illustrates the commendable growth of green coverage in China's built-up urban areas, which has exhibited a steady increase from 39.59 percent to 42.42 percent over the same ten-year period. These figures highlight the importance attributed to the establishment of green and high-quality cities. However, the creation of such cities goes beyond the mere consideration of the quantity and quality of green spaces; accessibility to green areas has emerged as a crucial criterion, as it determines the ease and promptness with which residents can access green spaces for recreational activities [7–9]. Nevertheless, there remains a dearth of rigorous quantitative analysis on the value of public and community-owned green spaces

in China. This research gap is also evident in Shanghai, renowned for its well-established real estate market and strategic commitment to sustainable urban development through the seamless integration of green construction practices [10–12]. Shanghai has earned prestigious designations such as the "Landscape Garden City", "International Garden City", and "National Garden City". Leveraging the availability of extensive apartment sales data in Shanghai, which can be combined with spatial information and community-specific characteristics, empirical analysis can be pursued. The rapid growth of China's real estate market has heightened the importance of various factors, including structural attributes, location characteristics, neighborhood features, and environmental considerations. Although these attributes cannot be traded in the same manner as conventional commodities, their impact on property values has been extensively studied using hedonic pricing models [13–15].

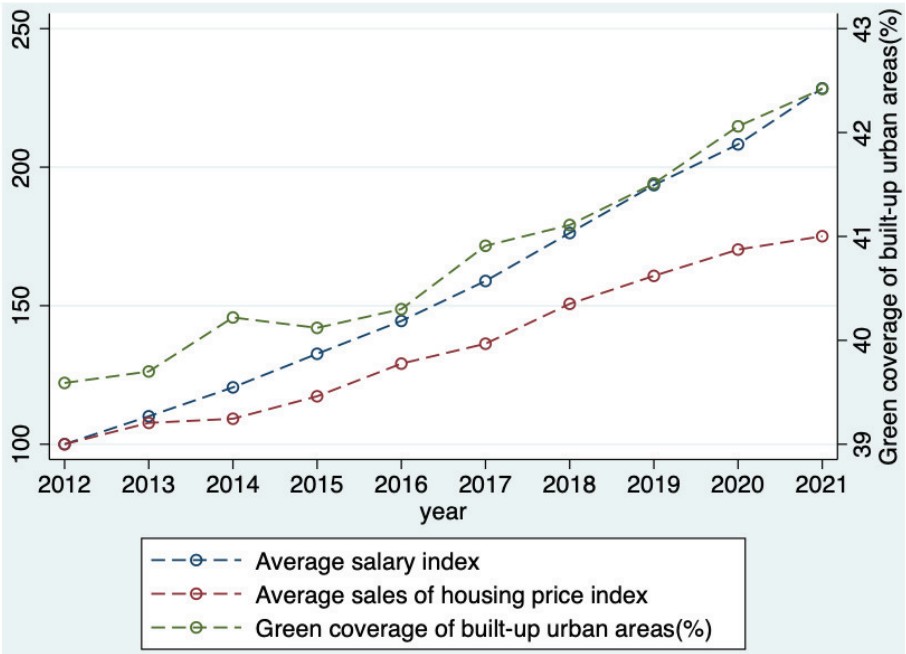

**Figure 1.** The long-term upward trend in the national average of income, housing price, and green space coverage. Data source: China Statistical Yearbook

This study employs a hedonic pricing model to empirically examine the relationship between the accessibility of public and community-owned green spaces and housing prices in Shanghai, China. Accurate and high-resolution data on green spaces were derived from the land-derived annual land cover product of China (CLCD), ensuring precision in the analysis. Community-level housing data were collected from the Soufang housing database, employing a crawler technique to gather information on community-owned green spaces, which goes beyond the scope of previous studies that primarily focus on public green spaces. Eventually, the comprehensive dataset encompassed 3388 communities across 138 subdistricts. The rich dataset addresses the potential endogeneity issue arising from the spatial dependence between green spaces and home values, an issue that has been largely overlooked in prior studies [3,11]. Surface water is naturally a compelling determinant for the spatial distribution of green spaces. The accessibility to water significantly influences the likelihood of green spaces being established in a local area. Moreover, the geographical distribution of surface water in a city renders it exogenous to current economic activities and residents' preferences. Consequently, we employ predetermined attributes as instrumental variables (IVs) for the endogenous green spaces, namely, the distance to the nearest water area and the ratio of surface water area to the total area.

In this paper, we carry out, to the best of our knowledge, the first comprehensive examination of the implicit value and spatial dependence of both public and community-owned green spaces and provide new empirical estimates in China's real estate market.

Our work contributes to the existing literature in multiple ways: (1) it introduces novel data that incorporate both quantity and quality indicators of green spaces; (2) to address spatial dependence and endogeneity concerns, the study utilizes a spatial hedonic pricing model and distinctive instrumental variables, e.g., the distance to the nearest water, the ratio of surface water area, and the geographical coordinates of communities; and (3) we consider the accessibility of both public and community-owned green spaces.

The paper is structured as follows. Section 2 reviews the relevant literature. Section 3 presents the hypotheses. Section 4 introduces the data and methodology. Empirical results are shown in Section 5, which is followed by some further analyses in Section 6. The last section concludes the paper.

## 2. Literature Review

Our paper relates to three strands of the literature: the relationship between housing sales prices and green spaces, studies on green spaces in China, and the hedonic pricing approach in empirical research.

### 2.1. Housing Price and Green Spaces

Urban green spaces have increasingly become a critical measure of livability for a city [16]. The multitude of social, ecological, and economic benefits of these spaces is widely acknowledged, including improving air quality, mitigating the urban heat island effect, enhancing aesthetic values [17], providing recreational facilities and opportunities, reducing noise pollution [18], enhancing the sense of community, and promoting physical and mental health [4,5,17,19,20]. Therefore, green spaces have become a crucial element in urban planning and design, providing many benefits to residents and contributing to the overall livability and sustainability of cities.

Green spaces are an important element of urban planning and design, contributing to environmental, social, and economic benefits. However, there exists no consensus on the direction and magnitude of the effect of green spaces among scholars. According to the classification of green spaces, recent studies have focused on the impact of specific types of green spaces, such as open spaces [17,21], parks [9,22], golf courses [23,24], greenbelt land [25,26], wetland [27], cemeteries [23,28], and so on. Despite these efforts, the research findings on the effects of these types of green spaces have generally been mixed, with positive, negative, and insignificant effects being variously reported for the same category of green spaces. The presence of open space amenities, such as public green spaces and natural parks, has been proven to impact the value of residential properties significantly. Using a geographic field model-based spatial hedonic method in Wuhan, China, proximity to the core area of Changjiang River recreation spaces and the East Lake scenic area has been found to increase the value of housing by 41.092 and 21.261 Yuan/m$^2$ for a 1% increase in the indices, respectively, [29]. Regarding parks, Wu et al. [9] classified the parks into forest, city, and community parks. The distance from a house to the nearest city park and community parks are positively related to housing price and result in 32.23% and $54, respectively.

Studies have shown that golf courses have a positive effect on residential house prices [23]. However, the impact of greenbelt land on housing prices appears to be mixed. Hui et al. [25] observed no influence of greenbelt proximity on apartment prices. Conversely, Gibbons et al. [26] reported that individuals were willing to pay more for houses situated in greenbelt locations that offered access to cities along with tight restrictions on housing supply.

Wetlands have positive and negative impacts on adjacent properties depending on their types [27]. Regarding cemeteries, their presence in the view of homes in Hong Kong tends to decrease property value due to the association of graveyards with ill fortune in traditional Chinese culture [28]. However, Lutzenhiser and Netusil [23] found that cemeteries, on average, do not have a statistically significant effect on housing prices.

### 2.2. Green Spaces in Various Countries

The relationship between green spaces and housing prices has been extensively studied in various regions, including the US [30], England [26], Korea [14], Portland [23], Denmark [13], and Hong Kong [17]. However, comparable studies focusing on China are still limited, leaving essential questions unanswered regarding how green spaces specifically impact housing prices in the country. Housing prices are influenced by a multitude of factors, with the availability, accessibility, and visibility of green spaces being particularly significant. Nevertheless, the specific attributes of green spaces that play a crucial role in determining the market value of residential properties remain unclear. As China has undergone rapid urbanization and industrialization in recent decades, it has faced unique environmental and social challenges. Therefore, a comprehensive study of green spaces in China holds particular importance in understanding their impact on housing markets and addressing the country's evolving urban landscape.

The availability of green spaces refers to the amount of green area within a specific distance from the residence of urban residents. This distance may vary depending on local context and preferences. Although green spaces may theoretically be available, we know nothing about their physical accessibility [8]. Recent studies have shown that the availability of green spaces can significantly impact housing prices, although these effects can vary across different districts. For example, by examining 545 residential communities in Ningbo, China, Liang et al. [31] found that the availability of urban public services had a significant effect on housing prices, although the effect was not uniform across all districts. Similar conclusions were reached by Hui et al. [25] in Hong Kong.

Green space-related variables included visual contact with green spaces, i.e., visibility [11,32]. In urban planning and environmental economics, visibility measures whether people can directly enjoy landscapes while in a house [17]. In the central built-up area of the Shenzhen Special Economic Zone (SSEZ), Chen and Jim [33] found that visual contact with the landscape was more highly valued than accessibility to green spaces. Recently, Wu et al. [34] investigated the capitalization effects of visual contact with green spaces on housing prices in Shenzhen, China, indicating that residents are willing to pay a premium for visual contact with outside green space views (39.5%) and inside green space views (17.7%).

Accessibility, which refers to the ease with which local amenities can be reached from a property [7], has gained significant importance in both academic and industry settings. The housing market reflects the willingness to pay a premium for proximity to green spaces, mainly to access the benefits of improved air quality and increased recreational activities [23]. In a study conducted in Shenzhen, China, Wu et al. [9] examined the accessibility of various green spaces, specifically community-owned parks and city parks. The results demonstrated a significant positive effect of accessibility on the surrounding real estate, with a premium of 54% for community-owned parks and 32.23% for city parks. Similar results have been reported for plazas and parks [33].

In summary, despite this growing attention to accessibility, there is a lack of literature investigating the relationship between green spaces, especially the availability of public and community-owned green spaces and housing prices in China. This gap underscores the need for further research to enhance the surrounding real estate values in urban contexts.

### 2.3. Hedonic Pricing Approach

Public goods, unlike regular commodities, can not be priced due to the lack of a formal market for their trade. Tiebout [35] conducted an equilibrium analysis, determining that household decisions regarding residential location are influenced by preferences and income. Consequently, the differences in the quality and quantity of public goods become capitalized in the surrounding housing price, making it a crucial factor for spatial disparities in residential value.

The hedonic pricing model, which is widely used in economics and other social sciences [36], has its roots in the utility theory proposed by Lancaster [37]. This theory posits that the utility derived from a good is not based on the good itself but rather on

the individual characteristics of the good. Rosen [38] presented the theory of hedonic pricing and provided a unified rendition to model implicit markets, arguing that the value of an item can be attributed to its unique characteristics. Building upon the theoretical foundation established by Harrison and Rubinfeld [39], the hedonic pricing model has become increasingly popular for estimating the value of public goods, medical care [40], automotives [41], and tourism [42].

The hedonic pricing model has a widespread application in assessing the contribution of different factors to housing prices [14,43,44]. In the existing literature, housing characteristics can be generally divided into three types: structure attributes, location attributes, and environmental attributes. Structure attributes refer to the physical characteristics of the property, such as size [22,45], orientation [46], decoration [47], floor [48], age [45], and the number of rooms [49]. Location attributes relate to the geographical features of the properties' surroundings, such as proximity to schools [30], public transportation [50], income level [15,47], school district [51,52], and unemployment rate [53]. Lastly, environmental attributes pertain to the ecological and environmental characteristics of the neighborhood, such as air quality [1,54] and green spaces [13–15].

## 3. Theoretical Hypotheses

Given the summary of related studies discussed above, we aim to explore the value of being close to urban green spaces in China under a spatial assessment framework. To this end, we propose several theoretical hypotheses in this section to guide the empirical strategies as follows:

### 3.1. Spatial Dependence in Residential Property Values

Economic activities or behaviors are based on specific locations so that they are interacted with and correlated spatially [2]. This spatial dependence on housing prices can be attributed to the accessibility of houses to one another, with the strength of the relationship diminishing as the distance between houses increases. This phenomenon is known as the adjacency effect in housing market literature [55]. The accessibility of homes often results in similar structural characteristics [56], shared location amenities, and social demographic characteristics [29,57], consequently, having comparable prices. This observation is further supported by the practical experience of real estate appraisers, who routinely take into consideration the value of neighboring properties when conducting their assessments [57].

On top of the empirical findings, the First Law of Geography formulated by Tobler [58] posits that there is a correlation between spatial distributions of geographical attributes, with objects in close accessibility being more related to each other than those located farther apart. Thus, built upon this principle, it is reasonable to hypothesize that spatial autocorrelation exists not only in house prices but also in green spaces and other housing attributes. To substantiate the analysis and conjecture, we first propose the following **hypothesis (H1)**:

**Hypothesis 1.** *There exist spatial interactions of residential property values and some home value determinants in the housing market.*

### 3.2. Property Value and Public Green Spaces

The manifold advantages of green spaces encompass social [19,20], ecological [17], and economic realms [14,43,44] and have gained widespread recognition within scholarly discourse.

Shanghai, a prominent global city in China, has strategically prioritized sustainable urban development by integrating green construction practices. Renowned as an exemplar of environmental stewardship, the city has garnered esteemed titles such as the "Landscape Garden City", "International Garden City", and "National Garden City". Analyzing comprehensive data extracted from the Shanghai statistical yearbooks spanning 2011 to 2021, an expansion in publicly accessible "park green spaces" (the public green spaces in this paper) dedicated primarily to recreational purposes becomes evident. Over a decade, public

green spaces have witnessed an impressive increase from 16,446 hectares to 21,981 hectares, representing a notable growth rate of 33.66%. These praiseworthy achievements stem from the collective endeavors of both public and private entities.

Extensive empirical evidence consistently highlights the substantial impact of public green spaces on the valuation of residential properties [9,29]. Expanding upon this body of research and aiming to corroborate the analysis and conjecture, the present paper presents the following **hypothesis (H2)**:

**Hypothesis 2.** *A higher degree of access to public green spaces around a community could raise the property value of housing units within the community.*

### 3.3. Property Value and Community-Owned Green Spaces

In China's urban areas, a substantial proportion of housing developments since the opening up of the housing market in the 1990s have adopted a residential club model. In this model, private developers establish civic amenities and services within the community, funded through residents' payments and managed by resident associations. As a result, an indispensable form of green spaces known as community-owned green spaces has emerged in China [10]. This unique attribute plays a crucial role in providing recreational and environmental benefits within these residential communities, fostering a sense of belonging, and enhancing residents' overall quality of life.

Community-owned green spaces play a vital role in urban infrastructure. From a physical perspective, these green spaces offer convenient opportunities for recreational activities, exercise, and social interaction within the local neighborhood [11,17,45]. These developments emphasize the significance of shared lifestyles and values within the "community" narrative, fostering increased social interaction among residents. However, their exclusive nature, enforced through physical barriers and controlled access, simultaneously excludes non-members from engaging in these social dynamics [59]. As a result, they directly contribute to enhancing the overall quality of life and daily living standards of residents.

Moreover, on a psychological level, community-owned green spaces provide aesthetically pleasing natural landscapes, which are often limited in urban settings [17]. They afford nearby residents the opportunity to enjoy visually appealing views, offering mental relief and a welcome escape from the monotonous urban architecture. The presence of these green spaces not only adds beauty to the surroundings but also exerts a positive influence on the well-being and mental health of residents.

In densely populated neighborhoods such as Shanghai, unlike western countries, the limited availability of outdoor areas has led to a noticeable absence of green spaces. Consequently, there is a strong inclination to compensate for this by utilizing public open spaces. Community-owned green spaces assume a pivotal role as valuable communal areas that allow individuals to embrace the outdoors and immerse themselves in natural environments. As a result, residents are willing to pay a premium for direct access to these community-owned green spaces. To examine the value of the community-owned product, we propose the following **hypothesis (H3)**:

**Hypothesis 3.** *If a residential community has a higher coverage ratio of green spaces, the apartments in it could be sold at higher prices.*

## 4. Data and Methodology

This section introduces the data we adopt to conduct empirical estimations from various sources, including both community and subdistrict-level attributes. Then, we outline the econometric framework and related empirical strategies.

*4.1. Data*

4.1.1. Geographical Information in Shanghai

We first introduce the geographical information used to process the GIS-based data in our cross-sectional analysis. According to Shanghai Civil Affairs Bureau [1], as of 2020, there are a total of 225 township-level divisions, i.e., subdistricts, including towns (*Zhen*), townships, and subdistricts (*Jiedao*) in Shanghai, of which our sample covers 138 subdistricts, as shown in Figure 2.

On top of the geographical boundaries at both district and subdistrict levels, we collected other geographical attributes that describe the relative distance between points of interest, such as public green spaces and communities. We gain the geo-data of ring roads, Huangpu River, and geographic locations of residential communities and map them into Figure 3. It displays that there exist three categories of location-based ring roads: inner-ring highways, middle-ring highways, and outer-ring highways, based on which all communities could be put into a range within the urban area. Another natural feature in Figure 3 is the Huangpu River, a prominent waterway located in Shanghai, China. The river plays a significant role in history, economy, and culture, as well as the geographical distribution of water areas and green spaces in Shanghai. The green dots represent the geo-locations of residential communities. Given the specific addresses of all communities, we geocode the locations. It is observed that most communities lie in the urban center and that we have sufficient sample units in each category of ring roads. Thus, it is important to control for location-specific effects related to both administrative levels and ring roads.

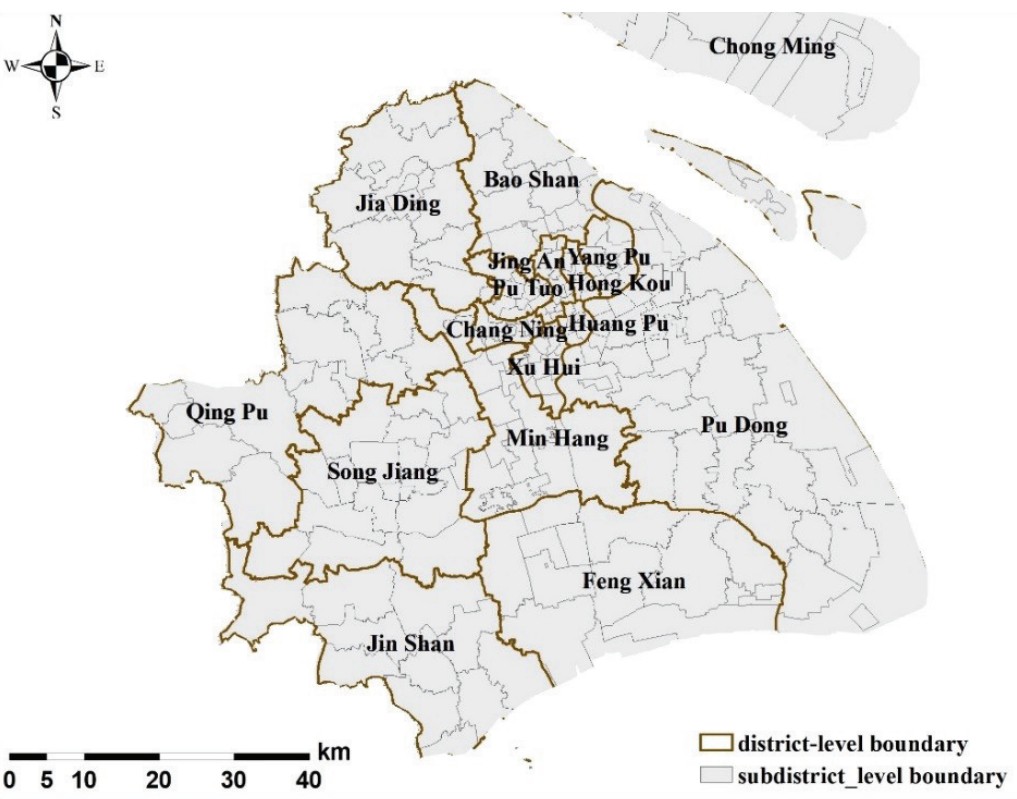

**Figure 2.** Geographical boundaries in Shanghai. This figure contains a total of 225 township-level divisions in Shanghai, of which our sample is over 138 subdistricts.

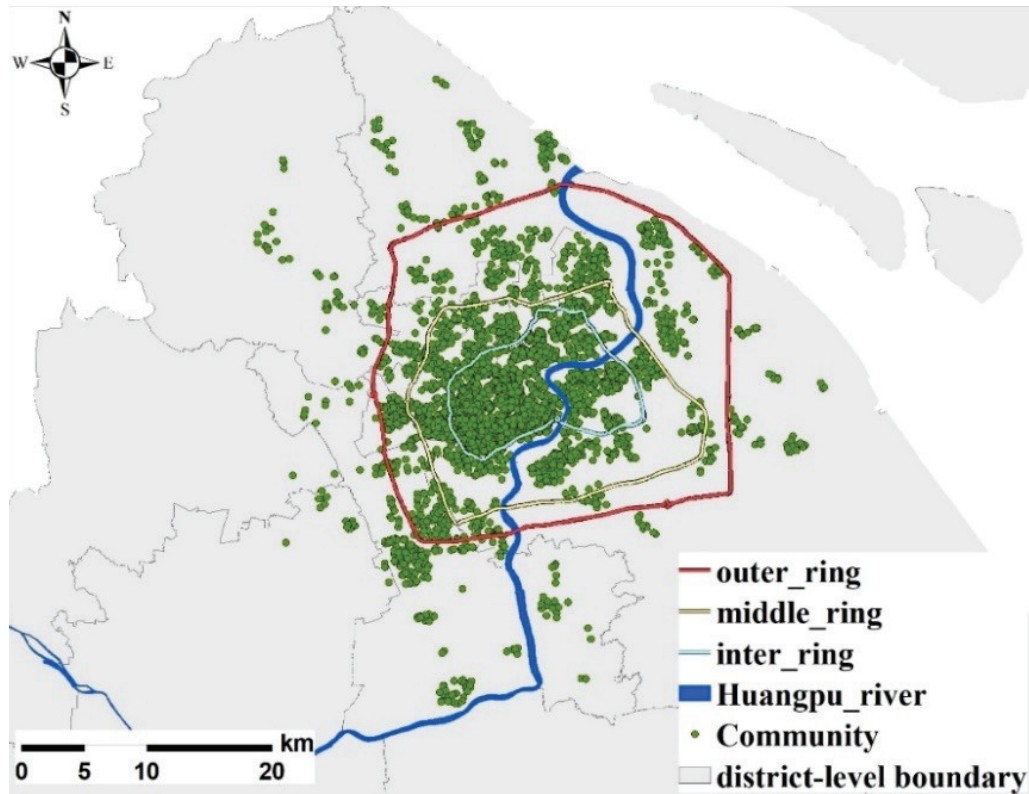

**Figure 3.** Ring roads, Huangpu river, and communities in Shanghai. This figure presents green spaces, geo-data of ring roads, Huangpu River, and geographic locations of residential communities.

4.1.2. Public Urban Green Spaces (PUGS)

To gain accurate and high-resolution data on urban green spaces, we adopt a Landsat-derived annual land cover product of China (CLCD) that provides 30-m annual land cover datasets using satellite-based images on the Google Earth Engine [2]. The CLCD dataset reveals the trends and patterns of land cover changes in China from 1985 to 2020, highlighting the impact of human activities and their dynamics on regional land surface cover under climate change in China [60]. Using high-resolution Landsat images data and standard Essential Urban Land Use Categories (EULUC) adapted from the Chinese Standard of Land Use Classification, we collect accurate geographical information related to urban green spaces in Shanghai [3]. Figure 4 below illustrates the geographical distribution of urban green spaces in 2020. It shows that most urban green spaces are in suburban areas, whereas few and smaller parcels of public green spaces are provided to residents living in the city center. Moreover, the green spaces are relatively evenly spread in terms of the distance to the urban center.

In this paper, we consider the number of green spaces, area of accessible green spaces, and distance to the nearest green spaces as three separate measurements of accessibility to green spaces. We set the range limit to 500 m, which is usually considered the maximum walking distance for a resident [4]. Given the specific locations of each parcel of green spaces and communities, we utilize the three separate indicators to quantify access to public green spaces.

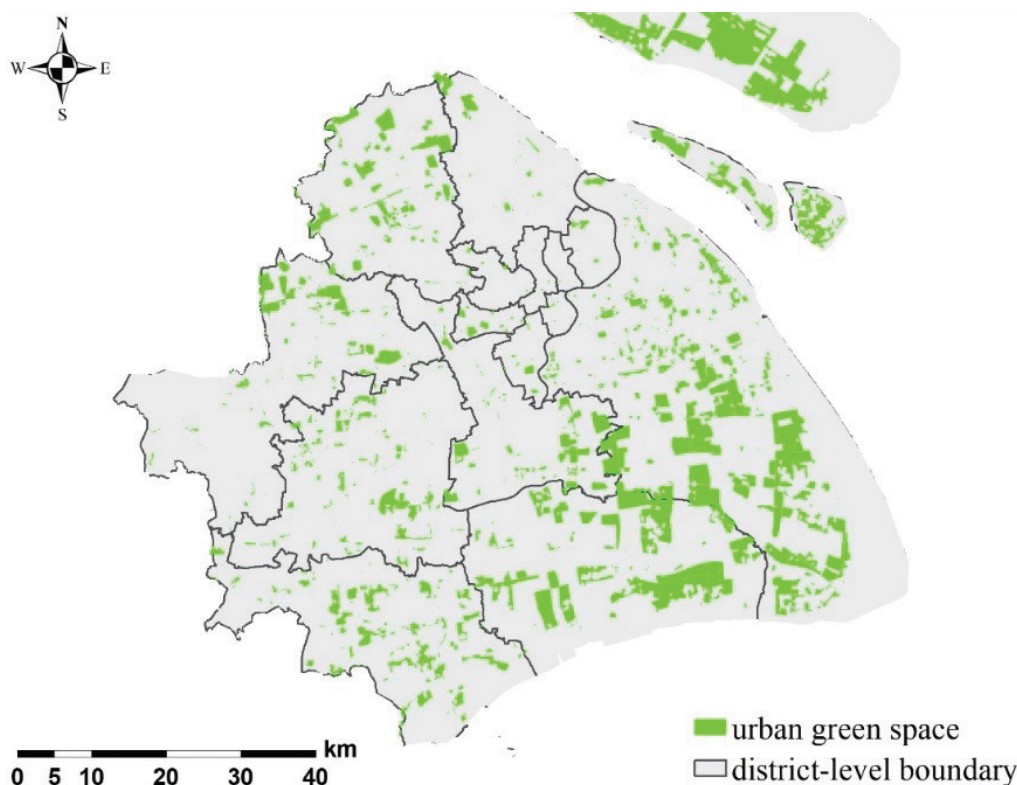

**Figure 4.** Geographical distribution of urban green spaces. This figure illustrates the geographical distribution of urban green spaces in 2020. Source: CLCD dataset

On top of individual access to green spaces, in order to more comprehensively evaluate the impact of green spaces on the value of housing services, we construct an overall index of green spaces accessibility used in our empirical analysis. Following Wu et al. [61], we develop a comprehensive floating catchment area (CFCA) to measure accessibility to urban green spaces [5]. The method of CFCA describes the attractiveness of public green spaces, e.g., parks, based on the number of amenities and the size of parks with different weighting schemes in a range. We first estimate the total demand for each parcel of green spaces given the total population within a catchment area proxied by the total number of housing units in all nearby communities. Then, we measure the relative attraction coefficient of green spaces, which is assumed to be influenced by size. Thirdly, we compute the green-to-population ratio, which is computed by dividing the attraction value by the total population within the catchment area. Lastly, given the range limit, a distance decay function, and green-to-population ratio, the spatial accessibility at every geographic unit is measured by the green-to-population ratio weighted by the distance that separates the residential zones from the urban green spaces. The computational procedure of the green spaces accessibility index is introduced in detail in Appendix A. All the data are computed by the Network Analyst tool in ArcGIS 10.8 [6].

Figure 5 presents the spatial distribution of the number of green spaces, area of accessible green spaces, distance to the nearest green spaces, and green spaces accessibility index of all communities. It shows in panel a that more green spaces are attainable for residents living in the suburb than in the urban center, whereas areas of green spaces accessible to residents are more evenly distributed. Panel c displays that the average distance of communities in the urban center to the nearest green spaces is shorter than that of ones far away from the urban center. The green spaces accessibility index is shown to be higher in the urban center due mainly to the shorter distance and larger population of residents covered by each parcel there.

As robustness checks, we construct the accessibility index with all nearby green spaces and green spaces greater than 100 square meters, as plotted in Figure A1 in Appendix C.

The empirical findings suggest that results are similar because most green spaces are greater than this lower bound.

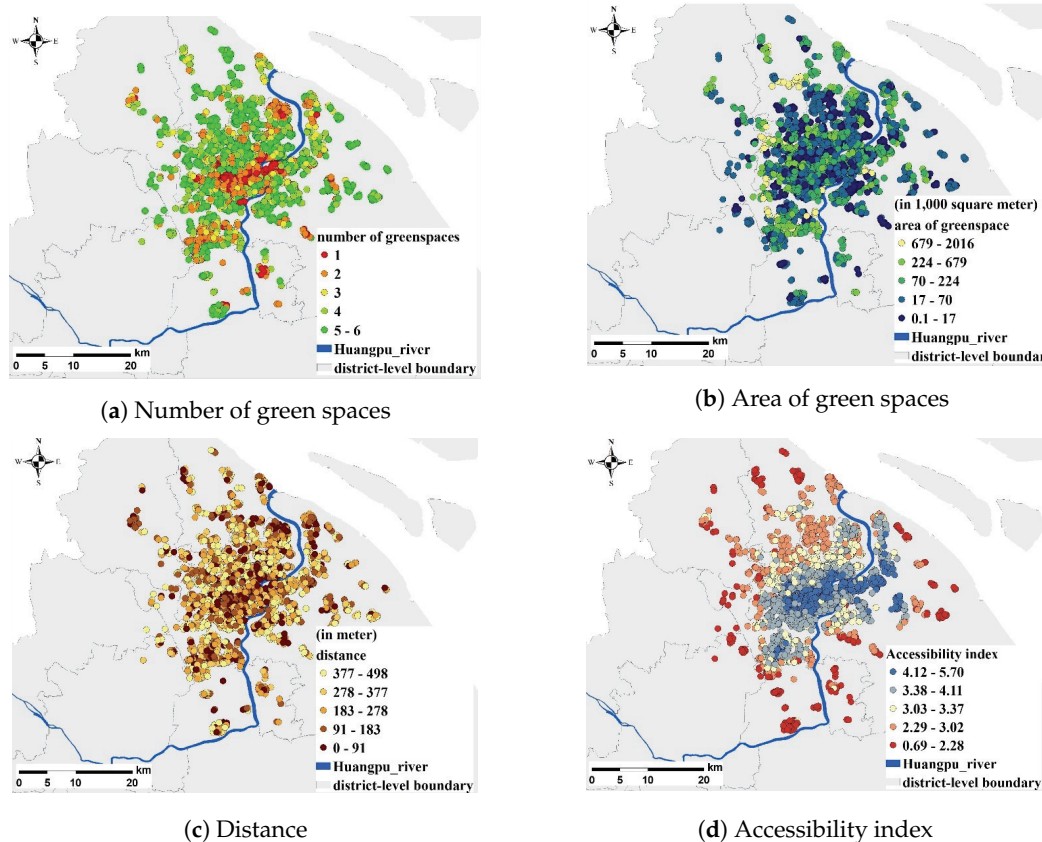

(**a**) Number of green spaces          (**b**) Area of green spaces

(**c**) Distance              (**d**) Accessibility index

**Figure 5.** Geographic distribution of green space–related variables in Shanghai. This figure presents the spatial distribution of the number of green spaces, area of accessible green spaces, distance to the nearest green spaces, and green space accessibility index of all communities in Shanghai.

### 4.1.3. Community-Level Housing Data

In addition to data on geographical boundaries and public urban green spaces (PUGS), we also collect housing data at the community level from the Soufang housing database, which is a prominent online real estate platform in China [7]. We utilized a crawler technique to derive the residential community data, including average sales price, the floor area ratio, green coverage rate, the total number of households, the year when the community was built, access to a heating system, and the specific address of the community. The green coverage rate refers to the ratio of the sum of the vertically projected area of greenery to the total land area of a residential area [8]. The floor area ratio (FAR) is the ratio of a building's total floor area (gross floor area) to the size of the piece of land upon which it is built. The accurate addresses are then geocoded to gain the longitude and latitude, which are later used to calculate the distance-related variables.

### 4.1.4. Subdistrict-Level Attributes

Given the geographical information of administrative boundaries, we also collected data on location-related attributes. In Shanghai, there exist 16 district-level administrative areas, below which there are subdistrict-level units. We obtained the data on areas of township-level subdistricts and surface water to compute the ratios of surface water area to total areas, which is used as an instrument for endogenous green spaces. Figure A2 in Appendix C shows the spatial distribution of water area and water areas ratios at the subdistrict level.

## 4.2. Data Matching and Summary Statistics

Given the data on geographical divisions, urban green spaces, community-level housing data, and subdistrict-level attributes, we merge and match them into a unique dataset that comprehensively describes the features of housing and urban green spaces in 2020 in Shanghai, China. This dataset could be adopted to estimate the value of the accessibility to green spaces in the context of the housing market in a spatial hedonic analysis.

Table 1 presents specific data sources and descriptive statistics for key variables incorporated into our empirical estimations. After cleaning the data and dropping observations with incomplete information, we gain the information of 3388 communities on the Soufang website in our sample in 2020. The administrative boundaries and ratios of water areas at the subdistrict level are presented in Figure 2. As of 2020, there are a total of 225 township-level divisions, named subdistricts, including towns (*Zhen*), townships, and subdistricts (*Jiedao*) in Shanghai, among which our sample covers 138 subdistricts. On average, a residential community possesses 3.47 green spaces of 121,983 square meters, which is 229.7 m away. In 2020, the average sales price of housing units in Shanghai was 52,244 CNY/m$^2$, and 87% of them are 70-year leasehold properties. Around 93% of communities have access to a heating system. The geographical information on ring roads is plotted in Figure 3.

Moreover, the average ratio of surface water area to total area at the subdistrict level is 0.057, which implies that surface-level water area consists of 5.7% of the geographical area. Figure A2 in Appendix C exhibits the spatial distribution of water area and water areas ratios at the subdistrict level and demonstrates that most water areas are in the suburban areas in Shanghai [9]. It also shows that there exist large variations in green space accessibility across communities, which is proper for the cross-sectional analysis in our setup.

**Table 1.** Summary statistics of key variables.

| Variables | Description | N | Mean | Std. | Min | Max | Source |
|---|---|---|---|---|---|---|---|
| Panel A: Accessibility to public green spaces | | | | | | | |
| count | Number of green spaces (unit) | 3388 | 3.470 | 3.323 | 1 | 6 | CLCD |
| area | Area of accessible green spaces (1000 m$^2$) | 3388 | 121.98 | 222.11 | 0.1079 | 2016.07 | CLCD |
| distance | Distance to the nearest green spaces (meter) | 3388 | 229.7 | 131.0 | 0 | 498.3 | CLCD |
| access | Accessibility index of green spaces (/) | 3388 | 3.590 | 0.799 | 0.695 | 5.698 | CLCD |
| access_1hm | Accessibility index of green spaces $\geq$ 10,000 m$^2$ (/) | 3388 | 3.350 | 0.694 | 0.612 | 5.314 | CLCD |
| Panel B: Community-level attributes | | | | | | | |
| unitprice | Sales price per unit area (1000 CNY/m$^2$) | 3388 | 52.244 | 18.140 | 7.965 | 190.00 | Soufang |
| greenratio | Area of greenery/total residential area (/) | 3388 | 0.342 | 0.089 | 0.0255 | 0.800 | Soufang |
| FAR | Gross floor area/total land area (/) | 3388 | 2.202 | 1.087 | 0.100 | 9.200 | Soufang |
| yearbuilt | Year when the community was built (year) | 3388 | 1998 | 19.55 | 1085 | 2018 | Soufang |
| # of unit | Number of housing units (unit) | 3388 | 764.1 | 2939 | 10 | 163,201 | Soufang |
| heating | Access to a heating system (/) | 3388 | 0.930 | 0.256 | 0 | 1 | Soufang |
| dist_sub | Distance to subway station (meters) | 3388 | 580.1 | 676.4 | 0.333 | 6370 | Soufang |
| property | Term of leasehold property: 40/70 [1] (/) | 3388 | 0.870 | 0.337 | 0 | 1 | Soufang |
| lon | longitude (degree) | 3388 | 121.5 | 0.069 | 121.2 | 121.7 | GIS |
| lat | latitude (degree) | 3388 | 31.23 | 0.065 | 31.00 | 31.45 | GIS |
| ring | Locations based on ring roads (/) | | | | | | GIS |
| | Within inner-ring highways (/) | 3388 | 0.429 | 0.495 | 0 | 1 | GIS |
| | Between inner- and middle-ring highways (/) | 3388 | 0.240 | 0.427 | 0 | 1 | GIS |
| | Between middle- and outer-ring highways (/) | 3388 | 0.181 | 0.385 | 0 | 1 | GIS |
| | Outside the outer-ring highways (/) | 3388 | 0.149 | 0.356 | 0 | 1 | GIS |
| Panel C: Subdistrict-level attributes | | | | | | | |
| area_subdis | Area of subdistrict (km$^2$) | 225 | 34.87 | 43.15 | 1.070 | 272.3 | CLCD |
| waterarea | Area of surface water (km$^2$) | 225 | 2.580 | 6.884 | 0.0003 | 64.00 | CLCD |
| waterratio | Ratio of surface water area to total area (/) | 225 | 0.055 | 0.053 | 0 | 0.393 | CLCD |

Notes: This table presents summary statistics of key variables for the 3388 communities and 225 subdistricts in our sample in 2020. As of 2020, there are a total of 225 township-level divisions, named as subdistricts, including towns (*Zhen*), townships, and subdistricts (*Jiedao*) in Shanghai, among which our sample covers 138 subdistricts, as shown in Figure 2. [1] A value of 1 denotes 70-year leasehold property, and a value of 0 represents 40-year leasehold apartment. The geographical information on ring roads is plotted in Figure 3.

## 4.3. Methodology

In accordance with previous hedonic studies, our work utilizes the community-level housing microdata in Shanghai in 2020 to estimate the value of living close to urban green spaces under the framework that involves hedonic pricing and spatial dependence.

### 4.3.1. Baseline Model

We first estimate the effects of accessible green spaces, location-specific attributes, and community-based features on the value of residential housing using the baseline regression model as follows:

$$\ln y_{id} = \theta_0 + \theta_1 \ln Green_{id} + X_{id}\Gamma + \gamma_d + \varepsilon_{id} \tag{1}$$

where $\ln y_{id}$ denotes the natural log of home price per square meter (CNY/m$^2$) for the community $i$ in subdistrict $d$ sold. $Green_{id}$ represents the green space-related indicators, i.e., accessibility index (access) [8,61], number of green spaces (count) [45], area of accessible green spaces (area) [3,10], and distance to the nearest green spaces (distance) [3]. $X$ is a vector of community-level attributes and location-based variables. $\gamma_d$ denotes the location fixed effects that control for unobserved local attributes that influence both green spaces and housing prices. $\varepsilon_{id}$ is the error term. Given the comprehensive evaluation of accessibility to green spaces, we adopt the estimation with green spaces accessibility index first and focus on its coefficient in interpreting the hedonic value of green spaces.

### 4.3.2. Spatial Hedonic Pricing Model

Local housing markets and resulting property values are often found to be spatially correlated across localized markets [1,62]. To control for the potential spatial dependence in property values and unobserved local price determinants, we adopt the spatial autoregressive model (SAR) with autoregressive disturbances following the setup by Anselin [63] [10]:

$$\ln y = \alpha I_n + \rho W_1 \ln y + X\beta + \mu$$
$$\mu = \eta W_2\mu + \varepsilon; \varepsilon \sim N\left(0, \sigma^2 I_n\right) \tag{2}$$

where $\rho$ and $\eta$ denote the spatial autoregressive parameters of the dependent variable and error term, respectively. They control for the spatial interactions in property values and price determinants. The spatial weighting matrix, $W_1$ and $W_2$, which could be either identical or different, measure the existing spatial spillover effects of housing prices and unobserved attributes between nearby areas. The key component in the spatial hedonic pricing model is the spatial weighting matrix, $W$, that controls for local spillovers to neighboring observations and describes spatial influences between nearby communities and green spaces as follows:

$$W = \begin{bmatrix} 0 & w_{12} & \cdots & w_{1n} \\ w_{21} & 0 & \cdots & w_{2n} \\ \vdots & \vdots & \ddots & \vdots \\ w_{n1} & w_{n2} & \cdots & 0 \end{bmatrix} \tag{3}$$

where the diagonal elements of the matrix are set to zero because there exist no spillover effects on itself. To fully capture the spatial impacts of green spaces, housing, and location-related attributes in high-density residential areas, we use the most common weighting method, i.e., inverse-distance weights, defined as $w_{ij} = 1/d_{ij}$, where $d_{ij}$ is the distance between the geographical centroids of community $i$ and $j$. Given the elements $w_{ij}$, the symmetric weighting matrix is row-standardized such that $\Sigma j w_{ij} = 1$. After the row standardization, the weighting matrix becomes asymmetric and is then used in the spatial model. The vector of disturbance, $\varepsilon$, follows a joint normal distribution. Here, the vector of $X$ includes the green space-related indicators, community-level attributes, location-based variables, and subdistrict-level fixed effects.

### 4.3.3. Instrumental-Variable Approach

The variable of interest in our hedonic pricing model is the green spaces accessibility. To find an accurate and unbiased measurement of its implicit value, the potential endogeneity issue needs to be fixed because the spatial distribution of green spaces is possibly

correlated with spatial patterns in local economic activities or other natural amenities that are also associated with the nearby home values. Thus, we adopt some predetermined attributes as instrumental variables (IV) for the endogenous green spaces. Surface water is naturally a good candidate for the spatial distribution of green spaces because access to water is strongly related to how likely green spaces are planted in a local area. Moreover, the geographical distribution of surface water in a city was formed many years ago, making it exogenous to current economic activities and residents' preferences. Given the high-resolution satellite data, we compute the distance to the nearest water area for each parcel of green spaces and the ratio of surface water area to total area in each subdistrict. We take as valid IVs the two exogenous variables, as well as the geographical coordinates of communities, in an instrumental-variable approach and estimate the two-stage spatial autoregressive model in our empirical analyses.

## 5. Empirical Results

In this section, we present the empirical results of the baseline and spatial models with an instrumental-variable approach.

### 5.1. Main Results of Non-Spatial Models

5.1.1. Standard Hedonic Pricing Model

To examine the impacts of public urban green spaces on local property values, we first estimate the baseline model (1). Table 2 presents the estimation results of the standard hedonic pricing model with the full sample. It displays that controlling for community-level attributes and location-related fixed effects is important to improve the accuracy of model estimation, given the significant variations of coefficients across columns. Incorporating the location-fixed effects at the level of finer geographical units improves the goodness of fit for the model. Thus, here we mainly focus on the results in column (4) that arise from the regression results with a complete set of community-level attributes and subdistrict-level fixed effects.

Table 2 shows that home prices, on average, increase by 0.05% if the green spaces accessibility index rises by 1%. Therefore, we verify the second hypothesis that, holding other factors equal, a higher degree of access to public green spaces around a community raises property values in the community. Compared to the research of Chen [3] and Jim [17], we obtain similar estimates on the value of public green spaces. Moreover, the more localized green spaces within a community are found to be more valuable for residents, given the larger marginal effect of the ratio of green spaces on property values. A 1% increase in the green ratio within a community is estimated to increase home values by 0.46%, much larger than that for public green spaces outside the community. Thus, we could confirm the hypothesis that apartments are sold at higher prices if a residential community has a higher coverage ratio of green spaces. Our findings closely align with the community-owned green spaces valuation estimates presented in previous studies [9,10].

Other than the green spaces, we find the impact of community-level attributes that are consistent with our conventional wisdom. Having access to a heating system and being closer to a subway station could raise home prices. The housing type is also found to play an important role, with the real properties of 70-year leasehold, on average, more valuable than those of 40-year leasehold apartments. The geographical locations in terms of ring roads are found to be very predictive of housing prices, and specifically, being closer to urban centers brings more price premiums on real properties.

**Table 2.** The property values and green spaces in the hedonic pricing model.

| Variables | (1) | (2) | (3) | (4) |
|---|---|---|---|---|
| | | Dependent Variable: $lny_{id}$ Home Price per Square Meter (CNY/m$^2$) | | |
| ln(Green) | 0.259 *** (0.031) | 0.135 *** (0.024) | 0.119 *** (0.021) | 0.050 *** (0.014) |
| greenratio | | 0.408 *** (0.078) | 0.483 *** (0.067) | 0.460** (0.056) |
| yearbuilt | | $90.614 \times 10^{-6}$ (0.0003) | 0.000074 (0.0004) | 0.00026 (0.0004) |
| num_unit | | $20.973 \times 10^{-6}$ $(10.742 \times 10^{-6})$ | $30.653 \times 10^{-6}$ $(20.123 \times 10^{-6})$ | $20.811 \times 10^{-6}$ $(10.961 \times 10^{-6})$ |
| FAR | | $-0.001$ (0.006) | $-0.005$ (0.005) | $-0.003$ (0.006) |
| heating | | 0.060 *** (0.018) | 0.052 ** (0.017) | 0.043 *** (0.014) |
| property_type | | 0.067 ** (0.025) | 0.073 *** (0.016) | 0.068 *** (0.016) |
| dist_subway | | $-0.00003$ * (0.00001) | $-0.00003$ ** (0.00001) | $-0.00003$ *** $(90.732 \times 10^{-6})$ |
| Outer-middle ring | | 0.163 *** (0.048) | 0.122 *** (0.039) | 0.153 *** (0.008) |
| Middle-inner ring | | 0.231 *** (0.030) | 0.161 *** (0.036) | 0.299 *** (0.006) |
| Within inner ring | | 0.396 *** (0.030) | 0.309 *** (0.044) | 0.705 *** (0.016) |
| Constant | 9.937 *** (0.100) | 9.845 *** (0.759) | 9.651 *** (0.813) | 9.587 *** (0.901) |
| Observations | 3388 | 3388 | 3388 | 3388 |
| R-squared | 0.276 | 0.410 | 0.454 | 0.563 |
| District FE | N | N | Y | N |
| Subdistrict FE | N | N | N | Y |

Note: Robust standard errors in parentheses are clustered at the district/subdistrict level, *** $p < 0.01$, ** $p < 0.05$, * $p < 0.1$ The communities with 40-year leasehold and located out of the outer ring road are taken at the reference group.

### 5.1.2. Results with Instrumental Variables

Because the spatial density and accessibility of urban green spaces might be endogenously determined in the housing market due to unobserved local socioeconomic attributes, such as the urban facilities or heterogeneous preferences of city planners, we adopt an instrumental-variable approach to estimating the impact of green spaces using geographical information of surface water area. Table 3 present the estimation results in an IV approach using different chosen IVs. Specifically, column (2) presents the estimation results with the IV of *dist_water*, i.e., the distance to the nearest water area. Column (3) presents the estimation results with the IV of the interaction term between *dist_water* and *waterratio*, i.e., the proportion of surface water area at the subdistrict level. Column (4) further adds the geographical coordinates, i.e., the longitude and latitude of each community.

We find that our parameter estimates with an IV approach are quantitatively different from that with an OLS approach, implying that the issue of endogeneity has led to a substantially underestimated influence of green spaces accessibility on property values. In column (4) with multiple IVs, we find that home prices, on average, increase by 0.165% if the green spaces accessibility index rises by 1%, more than three times as much in an OLS estimate. The Kleibergen–Paap rk LM statistics show the models are not under-identified. Moreover, the Kleibergen–Paap F-stats are greater than the relevant Stock–Yogo weak instrument test critical values (16.38 for a size distortion of a maximum of 10%) across all the specifications of columns (2–4). The Sargan–Hansen J statistic in column (4) suggests that the instruments are valid. After testing the validity of our instruments, we use the Durbin–Wu–Hausman test and find that green spaces accessibility is indeed endogenous and thus needs to be instrumented.

**Table 3.** The property values and green spaces in an IV approach.

| | | Dependent Variable: $lny_{id}$ Home Price per Square Meter (CNY/m$^2$) | | |
| --- | --- | --- | --- | --- |
| Variables | (1) (OLS) | (2) (IV) | (3) (IV) | (4) (IV) |
| ln(Green) | 0.050 *** (0.014) | 0.135 *** (0.023) | 0.220 *** (0.022) | 0.165 *** (0.038) |
| Observations | 3388 | 3388 | 3388 | 3388 |
| Communities-level attributes | Y | Y | Y | Y |
| Subdistrict FE | Y | Y | Y | Y |
| R-squared | 00.563 | 00.539 | 00.543 | 00.554 |
| Chosen IVs | | dist_water | waterratio * dist _ water | waterratio * dis_water + lon +lat |
| Kleibergen-Paap rk LM statistic | | 20.981 | 30.228 | 40.584 |
| (*p*-value) | | (0.041) | (0.035) | (0.022) |
| Kleibergen-Paap rk Wald F statistic | | 250.204 | 280.324 | 230.921 |
| Stock-Yogo critical values: | | | | |
| 10% maximal IV size | | 160.38 | 160.38 | 220.30 |
| 15% maximal IV size | | 80.96 | 80.96 | 120.83 |
| Sargan-Hansen J statistic | | | | 0.841 |
| (*p*-value) | | | | (0.655) |
| Durbin-Wu-Hausman test | | 23.274 | 26.859 | 15.689 |
| (*p*-value) | | (0.004) | (0.002) | (0.006) |

Note: robust standard errors in parentheses are clustered at the subdistrict level, *** $p < 0.01$. The column (2) presents the estimation results with the IV of dist_water. The column (3) presents the estimation results with the IV of the interaction term between waterratio and dist_water. The column (4) adds the geographical coordinates, i.e., longitude and latitude. dist_water denotes the distance to the nearest water area, while waterratio represents the proportion of surface water area at the subdistrict level. The Kleibergen–Paap rk LM statistic reports the underidentification test under the null hypothesis that the regression model is underidentified. We report critical values for the Kleibergen–Paap rk Wald F-statistics based on the Stock–Yogo weak instrument test with a size distortion of maximum 10% and a size distortion of maximum 15%, respectively. The Sargan–Hansen J statistic is used for testing over-identifying restrictions under the joint null hypothesis that the instruments are valid, i.e., uncorrelated with the error term. The Durbin–Wu–Hausman test reports the test of exogeneity of instrumented variables under the null hypothesis of exogeneity.

## 5.2. Main Results of Spatial Hedonic Models

### 5.2.1. Analysis of Spatial Dependence

We first examine the potential spatial dependence of our key variables and residuals of main regression models using the Global Moran *I*'s test of spatial dependence. Table 4 displays that both property values and green spaces are found to be spatially distributed at the 1% level. Moreover, the error terms in the hedonic pricing models estimated by OLS and IV approaches are also spatially correlated. Thus, we verify the existence of spatial interactions of residential property values and some home value determinants in the housing market.

**Table 4.** Moran *I*'s test of the spatial Dependence.

| | Home Value | Green Spaces | OLS | IV |
| --- | --- | --- | --- | --- |
| Moran *I* test statistic | 25,783.28 *** | 60,170.15 *** | 14.471 *** | 12.165 *** |
| (*p*-value) | (0.000) | (0.000) | (0.000) | (0.000) |
| Observations | 3388 | 3388 | 3388 | 3388 |

Note: *p* values are reported in parentheses. *** $p < 0.01$.

### 5.2.2. Spatial Hedonic Model with Instrumental Variables

Due to the existence of spatial dependence in green spaces, home values, and other price determinants, we estimate the spatial hedonic pricing model with both OLS and IV approaches, as shown in Table 5. The column (1) exhibits the IV estimation results

without controlling for the spatial dependence. The column (2) presents the estimation results of a spatial autoregressive model with an OLS estimation. The column (3) presents the estimation results of a spatial autoregressive model with an IV estimation. Overall, it unveils the necessity of incorporating spatial dependence in property values and error terms. We could find in the estimated results of the SAR-OLS and SAR-IV models that home values are positively correlated across nearby areas, whereas the unobserved price determinants are significantly and negatively impacted by surrounding areas, even though the influence is not statistically significant.

Table 5 also displays the direct, indirect, and total effects of green spaces accessibility on home values. The average own-community direct effect of a 1% increase in urban green spaces accessibility is to raise the home value by 0.13%. The across-community spillover effect of a 1% increase in green spaces accessibility is to raise the average property value in a nearby community by 0.082%. The total effects are the sum of the direct and indirect effects, which suggests that the higher degree of urban green spaces accessibility improves the overall desirability of the residential apartments. As robustness checks, we also present the estimation result with the accessibility index constructed by green spaces greater than 100 square meters. Table A1 in Appendix B demonstrates that the empirical results are robust to the selected parcels of green spaces.

**Table 5.** Spatial hedonic pricing model in an IV approach with an overall index.

| | Dependent Variable: $lny_{id}$ Home Price per Square Meter (CNY/m$^2$) | | |
|---|---|---|---|
| **Variables** | **(1)** **Non-Spatial IV** | **(2)** **SAR-OLS** | **(3)** **SAR-IV** |
| ln (Green) | 0.165 *** (0.013) | 0.246 *** (0.014) | 0.207 *** (0.019) |
| $Wlny$ | | 0.570 *** (0.119) | 0.567 *** (0.123) |
| $W\mu$ | | −0.287 (0.256) | −0.287 (0.257) |
| Observations | 3388 | 3388 | 3388 |
| Communities-level attributes | Y | Y | Y |
| Subdistrict FE | Y | Y | Y |
| $R^2$/Pseudo $R^2$ | 0.554 | 0.113 | 0.033 |
| | direct | indirect | total |
| Marginal effects of SAR-IV | 0.131 *** (0.024) | 0.082 *** (0.038) | 0.213 *** (0.070) |

Note: robust standard errors in parentheses are clustered at the subdistrict level, *** $p < 0.01$. Column (2) presents the estimation results of a spatial autoregressive model with an OLS estimation. Column (3) presents the estimation results of a spatial autoregressive model with an IV estimation. Delta-method standard errors are reported in parentheses at the bottom in computing the marginal effects in the SAR-IV model.

## 6. Further Analyses

### 6.1. Analysis of Individual Indicators

On top of the analysis of the overall green spaces accessibility, we decompose the index and analyze the three individual indicators that influence the home values separately in the SAR-IV model. The three individual measurements are the number of green spaces, the area of accessible green spaces, and the distance to the nearest green spaces, which, from different aspects, reflects how accessible the local green spaces are for households living nearby. Table 6 presents the estimation results, showing that property values in a community could be related to each of the three indicators. Specifically, a residential community that accesses more and larger green spaces within a shorter distance is supposed

to be sold at a higher price. The causal relationship proves very robust and significant across columns.

### 6.2. Heterogeneity Analysis

The revealed hedonic value of green spaces accessibility might vary by some socioeconomic variables, such as household preferences, income, or local public transport. In addition to the global estimation, we aim to find the heterogeneity in the price premiums of green spaces. Because the overall green space accessibility index could reflect and provide more information on its overall impact on property values, we focus on the overall index in our heterogeneity analysis.

Given the data availability, we first explore the location-based heterogeneous effects of accessibility to public green spaces on property values. Here we analyze the subgroups of communities based on the ring roads. As shown in Table 7, generally, the home buyers living closer to the urban center value the accessibility of public urban green spaces more than those purchasing apartments far away from the urban center. It implies that the scarcity in urban areas increases the value of green spaces because a majority of developable lands are used to construct commercial buildings rather than urban amenities, e.g., public parks or greenways. We also explore the heterogeneity in property value and housing type and demonstrate: (1) that access to green spaces for more valuable homes, which are more likely to be purchased by higher-income households, is given a higher price premium and (2) that landlords of 70-year leasehold apartments are willing to pay more for the accessibility for green spaces than 40-year leasehold housing units, which are relatively cheaper. The heterogeneity analysis reveals that, even within an urban city, the willingness to pay (WTP) for the important urban amenity varies substantially by location, household income, and preferences. Thus, we need to consider the heterogeneity in our analysis when evaluating its actual value for local residents.

**Table 6.** Spatial hedonic pricing model in an IV approach with separate indicators.

| | Dependent Variable: $lny_{id}$ | | | |
| | Home Price per Square Meter (CNY/m$^2$) | | | |
| **Variables** | **(1)** SAR-IV | **(2)** SAR-IV | **(3)** SAR-IV | **(4)** SAR-IV |
|---|---|---|---|---|
| area_2020 | $6.943 \times 10^{-8}$ ** ($3.099 \times 10^{-8}$) | | | $5.813 \times 10^{-8}$ *** ($1.142 \times 10^{-8}$) |
| count_2020 | | 0.004 ** (0.002) | | 0.004 ** (0.002) |
| dist_2020 | | | −0.0002 *** (0.000) | −0.0001 ** (0.000) |
| $Wlny$ | 0.495 *** (0.117) | 0.554 *** (0.113) | 0.574 *** (0.114) | 0.503 *** (0.118) |
| $W\mu$ | −0.257 (0.258) | −0.293 (0.258) | −0.254 (0.254) | −0.239 (0.257) |
| Observations | 3388 | 3388 | 3388 | 3388 |
| Communities-level attributes | Y | Y | Y | Y |
| Subdistrict FE | Y | Y | Y | Y |
| Pseudo $R^2$ | 0.045 | 0.033 | 0.035 | 0.087 |

Note: robust standard errors in parentheses are clustered at the subdistrict level, *** $p < 0.01$, ** $p < 0.05$. Delta-method standard errors are reported in parentheses at the bottom in computing the marginal effects of the SAR-IV model.

**Table 7.** Heterogeneity analysis in the SARAR-IV model.

| | | | | |
|---|---|---|---|---|
| | Dependent Variable: $lny_{id}$ Home Price per Square Meter (CNY/m$^2$) | | | |
| | **Heterogeneity by Ring Roads** | | | |
| **Variables** | **(1)** **Outside the Outer Ring** | **(2)** **Bet Middle & Outer Ring** | **(3)** **Bet Middle & Inner Ring** | **(4)** **Within Inner Ring** |
| ln(Green) | 0.172 *** (0.015) | 0.206 *** (0.014) | 0.231 *** (0.019) | 0.277 *** (0.023) |
| $Wlny$ | 0.395 *** (0.127) | 0.534 *** (0.113) | 0.674 *** (0.121) | 0.703 *** (0.108) |
| $W\mu$ | −0.197 (0.318) | −0.293 (0.258) | −0.251 (0.254) | −0.212 (0.197) |
| Observations | 506 | 614 | 814 | 1454 |
| Communities-level attributes | Y | Y | Y | Y |
| Subdistrict FE | Y | Y | Y | Y |
| Pseudo $R^2$ | 00.025 | 00.023 | 00.025 | 00.027 |
| | **Heterogeneity by property value** | | **Heterogeneity by housing type** | |
| **Variables** | **(5)** **VALUE > 50%** | **(6)** **Value $\leqq$ 50%** | **(7)** **70-year leasehold** | **(8)** **40-year leasehold** |
| ln(Green) | 0.312 *** (0.011) | 0.206 *** (0.014) | 0.331 *** (0.021) | 0.212 *** (0.021) |
| $Wlny$ | 0.512 *** (0.097) | 0.334 *** (0.113) | 0.674 *** (0.121) | 0.433 *** (0.118) |
| $W\mu$ | −0.231 (0.281) | −0.233 (0.218) | −0.278 (0.241) | −0.223 (0.192) |
| Observations | 1694 | 1694 | 2946 | 442 |
| Communities-level attributes | Y | Y | Y | Y |
| Subdistrict FE | Y | Y | Y | Y |
| Pseudo $R^2$ | 0.031 | 0.031 | 0.035 | 0.031 |

Note: robust standard errors in parentheses are clustered at the subdistrict level, *** $p < 0.01$. Delta-method standard errors are reported in parentheses at the bottom in computing the marginal effects of the SAR-IV model.

## 7. Discussion and Conclusions

This study utilizes hedonic price and spatial econometric models, leveraging data from 3388 residential communities and 225 subdistricts in Shanghai, China in 2020, to quantitatively explore the relationship between property value and the accessibility of public and community-owned green spaces. The research findings yield significant insights: (1) a 1 % increase in the overall accessibility of green spaces is associated with an average increase in home prices of 0.17%, highlighting the positive impact of enhanced accessibility on property values, (2) a 1% increase in the green ratio within a community is associated with a significant 0.46% increase in property values; (3) the number of accessible green spaces, the area of accessible green spaces, and the distance to the nearest green spaces individually contribute to higher home values.

Understanding the hedonic value of green spaces plays a pivotal role in driving sustainable urban development, enhancing residents' quality of life, and recognizing the intrinsic worth and advantages of integrating green spaces into urban environments. From the perspective of real estate developers, the incorporation of green spaces into the design and development of new residential projects assumes paramount importance. The creation and preservation of accessible green spaces within and around communities significantly enhance the attractiveness and value of properties. Collaborative efforts with urban planners and landscape architects are essential to seamlessly integrate green spaces into housing developments, thereby contributing to the overall livability and appeal of the community. Developers can strategically leverage the distinctive landscape characteristics of their projects as compelling selling points during product positioning and market promotion to attract prospective consumers.

Moreover, this paper furnishes invaluable insights for stakeholders involved in urban construction. When deliberating on green spaces or the broader concept of green infrastructure, local governments should adopt a human-oriented principle, which entails respecting and accommodating the diverse demands of urban residents with varying socioeconomic characteristics. Additionally, governments should ensure equitable distribution of green resources, considering the social ramifications of policy implementation to guarantee that residents from all social strata have equitable access to ample green spaces [65,66]. In the realm of urban design, greater emphasis should be placed on facilitating convenient access to green spaces and promoting social equality, transcending a sole focus on economic value. For urban planners, the incorporation of sustainable urban design principles and green infrastructure planning in city development projects assumes critical importance. Prioritizing the integration of public and community-owned green spaces enhances the overall attractiveness and livability of the city.

Understanding the hedonic value associated with different types of green spaces paves the way for sustainable urban development, elevating residents' quality of life, and recognizing the intrinsic worth and advantages of integrating green spaces into urban environments. These proactive initiatives culminate in the creation of vibrant, sought-after, and livable communities that place the well-being and contentment of their residents at the forefront.

**Author Contributions:** S.B. carried out the conceptualization and methodology. H.Z. finished the data collection and processing. J.L. finished the original draft preparation and writing as well as funding acquisition. R.W. conducted the review and editing of the manuscript. All authors read and approved the final manuscript.

**Funding:** Authors acknowledge the financial support from Zhejiang Provincial Philosophy and Social Sciences Planning Project (23NDJC078YB) and the Research Fund by the Social Sciences Federation of Yuhang District in Hangzhou, Zhejiang (Yhsk23C09).

**Data Availability Statement:** Not applicable.

**Conflicts of Interest:** The authors declare no potential conflicts of interest with respect to the research, authorship, and publication of this article. This article does not contain any studies involving human participants performed by any of the authors.

## Appendix A. Data Construction of the Green Space Accessibility Index

In this section, we describe how we construct the index of green space accessibility used in our analysis, in order to more comprehensively evaluate the impact of green spaces on the value of housing services. Following Wu et al. [61], we develop a comprehensive floating catchment area (CFCA) to measure urban green spaces accessibility that describes the attractiveness of public green spaces, e.g., parks, based on the number of amenities and the size of parks with different weighting schemes in a range. The first advantage of CFCA is that catchments of varying sizes are adopted to reflect service ability because larger parks can serve more distant residents. Second, different modes of transportation, such as driving, biking, and walking, can be estimated in our proposed model. We introduce the CFCA method as follows

### Appendix A.1. Total Demand for Green Spaces

Residents may choose to not visit crowded urban green spaces. The size of the urban green spaces is divided by the total population within its catchment. The total demand, $D_j$, for each green spaces $j$ is calculated by summing the respective populations, $r_i$, in this set $G(d_{ij}, d_0)$ and using a distance decay function:

$$D_j = \sum_{i \in \{d_{ij} \leq d_0\}} r_i G(d_{ij}, d_0) \tag{A1}$$

where $r_i$ is the population in residential zone $i$, which falls into catchment ($d_{ij} \leq d_0$) for green spaces $j$. Here the total population within the catchment area arises from the total number of housing units in all nearby communities. $d_{ij}$ is the travel distance between residential zone $i$ and green spaces $j$, where $d_0$ is the threshold travel distance; and $G$ is the friction-of-distance listed below:

$$
G\left(d_{ij}, d_0\right) = \begin{cases} \dfrac{e^{-(1/2)*\left(d_{ij}/d_0\right)} - e^{-(1/2)}}{1 - e^{-(1/2)}}, & \text{if } d_{ij} \leq d_0 \\ 0, & \text{if } d_{ij} > d_0 \end{cases} . \tag{A2}
$$

*Appendix A.2. Relative Attraction Coefficient of Green Spaces*

Even if one could argue that maintenance and amenities make parks attractive, given the limited data, we rest on the assumption that park attraction is based on size alone. We measure the attraction coefficient influenced by the size:

$$
G_j = \frac{S_j}{\sum_j S_j} \tag{A3}
$$

where $S_j$ is the area of urban green spaces $j$, $G_j$ is the relative attraction coefficient of green spaces.

*Appendix A.3. Green-to-Population Ratio*

We use $P_j$ to present the green-to-population ratio, which is computed by dividing the attraction value by the total population within the catchment area:

$$
P_j = \frac{G_j}{\sum_j D_j} \tag{A4}
$$

*Appendix A.4. Accessibility Index of Urban Green Spaces*

A distance decay function $G\left(d_{ij}, d_0\right)$ gives a higher weight to the park-to-population ratio of an urban green spaces. Given the range limit, a distance decay function, and green-to-population ratio, the spatial accessibility at geographic unit $i$ is then defined as:

$$
A_i = \sum_{i \in \left\{d_{ij} \leq d_0\right\}} P_j G\left(d_{ij}, d_0\right) \tag{A5}
$$

where the spatial accessibility is measured by the green-to-population ratio weighted by the distance that separates the residential zones from the urban green spaces. The higher the accessibility score, the greater the accessibility of that residential zone to urban green space compared to all other residential zones in the study area.

All the data are computed by the Network Analyst tool in ArcGIS 10.8, which offers a robust and integrated platform for working with spatial data. As robustness checks, we construct the accessibility index with all nearby green spaces and green spaces greater than 100 square meters. As shown in the section above, the results are similar because most green spaces are greater than this lower bound.

## Appendix B. Additional Tables

**Table A1.** Spatial model in an IV approach with green spaces larger than 1 hm.

| Variables | (1) Non-Spatial IV | (2) SAR-OLS | (3) SAR-IV |
|---|---|---|---|
| | **Dependent Variable:** $lny_{id}$ **Home Price per Square Meter (CNY/m$^2$)** | | |
| ln(Green) (>1 hm) | 0.125 *** (0.011) | 0.221 *** (0.010) | 0.145 *** (0.013) |
| *Wlny* | | 0.412 *** (0.102) | 0.474 *** (0.131) |
| *Wμ* | | −0.223 (0.156) | −0.224 (0.217) |
| Observations | 3388 | 3388 | 3388 |
| Communities-level attributes | Y | Y | Y |
| Subdistrict FE | Y | Y | Y |
| Pseudo $R^2$ | 00.557 | 00.122 | 00.041 |
| Marginal effects of SAR-IV | direct 0.135 *** (0.022) | indirect 0.061 *** (0.024) | total 0.196 *** (0.067) |

Note: Robust standard errors in parentheses are clustered at the subdistrict level, *** $p < 0.01$. Column (2) presents the estimation results of a spatial autoregressive model with an OLS estimation. Column (3) presents the estimation results of a spatial autoregressive model with an IV estimation. Delta-method standard errors are reported in parentheses at the bottom in computing the marginal effects of the SAR-IV model.

## Appendix C. Additional Graphs

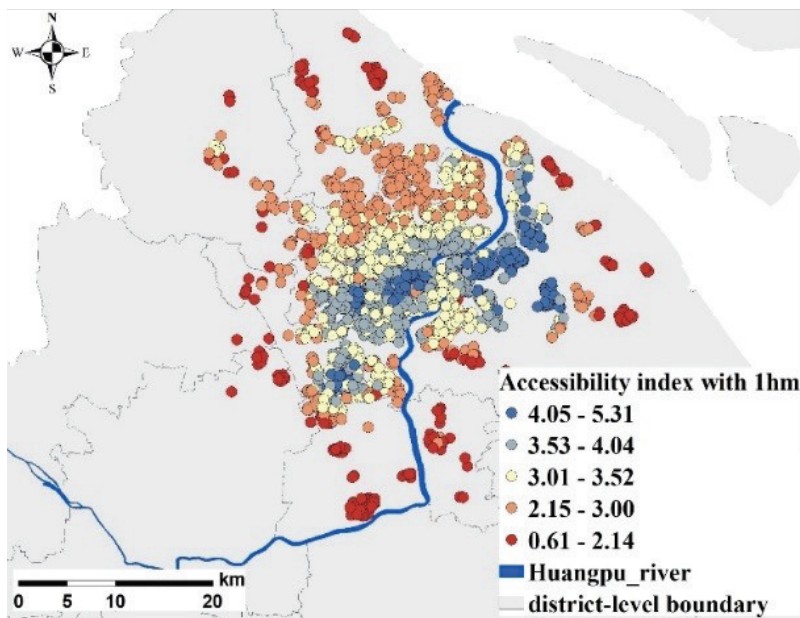

**Figure A1.** Spatial distribution of accessibility index with green spaces over 1 hm. This figure presents the accessibility index with all nearly green spaces and green spaces greater than 100 square meters.

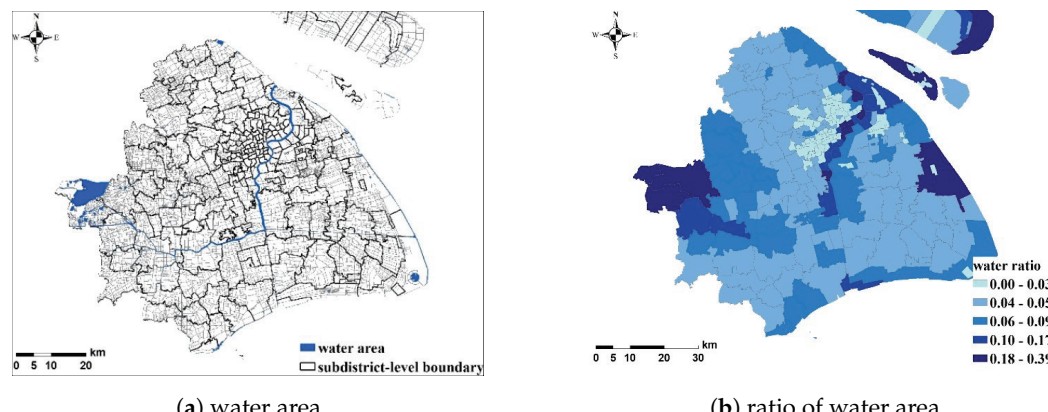

(**a**) water area                                    (**b**) ratio of water area

**Figure A2.** Spatial distribution of water area and water areas ratios at the subdistrict level.

## Notes

1. https://mzj.sh.gov.cn/ (accessed on 1 April 2023).

2. https://zenodo.org/record/5210928 (accessed on 3 May 2023).

3. According to standard Essential Urban Land Use Categories (EULUC), public green spaces (code 0505) include lands used for entertainments and environmental conservations, such as parks, trees planted in rows parks, special parks, scenic areas, urban wetlands, forest parks, nature reserves, and residential parks [60].

4. As a robustness check, we perform a stepwise increase of 100 m to the range limit of 1000 m and find no significant variations on the impact of green spaces.

5. The first advantage of CFCA is that catchments of varying sizes are adopted to reflect service ability because larger parks can serve more distant residents. Second, different modes of transportation, such as driving, biking, and walking, can be estimated in our proposed model.

6. This paper utilizes the ArcGIS 10.8, a powerful geographic information system software suite designed to analyze, visualize, and manage spatial data for various applications in fields such as geography, environmental science, urban planning, and more [8,34]. Given its comprehensive and integrated capabilities, we calculate all spatial statistics in this platform.

7. Soufang holding (NYSE: SFUN) is a publicly traded company listed in New York Stock Exchange. Fang.com is the website launched by the listed company that provides the real estate data in the housing market of Shanghai.

8. Based on the data provided by Soufang and the common standard, the green coverage rate is referred to as the ratio of the sum of the vertically projected area of greenery to the total land area of a residential area.

9. The locations of water areas are closely related to the spatial distribution of urban green spaces, which is used as an instrumental variable (IV) for endogenously determined green spaces. We specify the IV-approach in later section.

10. We estimate the SAR model using the ordinary least squares (OLS) because the estimator for spatial autoregressions remain consistent as pointed out by Lee (2002), as long as spatially lagged regressors are non-stochastic. In this case, the model can be simply estimated using the OLS procedure [64].

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
