# Peer review of "Valuing the Accessibility of Green Spaces in the Housing Market: A Spatial Hedonic Analysis in Shanghai, China"

_land, doi:10.3390/land12091660_

Round 1

Reviewer 1 Report

Introduction: The authors didn’t adequately justify the reasons to do this study. More most relevant and recent advances in the field should be presented.

Line59-67: It is a bit weird to mention the results in Introduction part.

Line68-69: Please explain why these variables were used.

2.2 Only talking about the situation in China is not enough for a paper published in an international journal. The general situation of the study on green spaces all around the world should be reviewed. The special situation in China can be highlighted in Introduction part.

2.1 and 2.2 have some repetitive content.

7 Discussion part is very weak as many of the findings have not been further explained or elaborated.

The English language should be improved. Grammar errors and typos should be corrected. For instance, Line 82, Line112.

Reviewer 2 Report

This study aims to quantify the effect of green spaces through spatial regression. The analysis result confirms the presence of the impact of green spaces and spatial correlation in residential property values. A wide variety of related studies are reviewed in this study. 

However, there are a large number of simular studies demonstrating the effect of green spaces although those in Shanghai might be relatively limited. Therefore, the novelty is relatively weak.

Besides, there are some parts revisions are needed in the methodology parts:

- Although the authors mention that an OLS approach is used to estimate the spatial autoregressive model (e.g., line 514), this model cannot be estimated by OLS. 

- The authors assume many explanatory variables in X including district-level fixed effects based on line 432, which can confound with the spatial autoregressive terms. In such a situation, multicollinearity can be severe. It is needed to check if the model suffer from multicollinearity.

Other minor comments are listed below:

- Line 105: What does "GFM" stand for?

- Line 187-188: Although the authors classified housing characteristics into four types, the four seem partly overlappying. For example, "location attributes" seem to include "neighborhood attributes" and "environmental attributes". "environmental attributes" can be throught as a sort of "neighborhood attributes".

- Equation (1): theta_0,theta_1, and Γ are not defined.

- Equation (1): Is Green_id a vector? Please clarify.

- Although the authors mentioned that "we use the most common weighting method, i.e., an inverse-distance weits" (line 426), it is not true. In my understandring, sparse W such as 4 nearest neighbors-based W or adjacency-based W are more popular because of the memory efficiency and numerical stability in spatial econometrics, which is the major area of spatial autoregressive model.

- Line 428: Please fix "such that Sigma_j w_ij" to "such that Sigma_j w_ij=1"

Reviewer 3 Report

Ensure that all references are consistently formatted. For instance, some references have a space between the author's initials and surname, while others don't.

Cross-check the validity of all DOI links provided to ensure they lead to the correct papers.

Table 1: Summary statistics of key variables (Page 11):

Consider adding units to all the variables for clarity. For instance, specify the unit for "Area of accessible greenspaces" (e.g., square meters, hectares).

The source for some data is listed as "CLCD" and "Soufang." Ensure that these sources are explained or referenced somewhere in the document.

The description for "ring" variable seems to be cut off. Ensure that all descriptions are complete.

Appendices (Page 21):

 The formulas in the appendices (e.g., A3, A4, A5) could benefit from a brief explanation or interpretation to make them more accessible to readers unfamiliar with the methodology.

Ensure that symbols like "Gj" and "Sj" are defined before they are used in formulas

Content Flow:

The content seems to jump between different sections (e.g., from references to tables to appendices). Ensure there's a logical flow to the document, with each section smoothly transitioning to the next.

Clarity and Detail:

 The section on "Accessibility index of urban green spaces" (Page 21) mentions using the Network Analyst tool in ArcGIS 10.8. It might be helpful to provide a brief overview or citation about this tool for readers unfamiliar with it.

The document mentions "greenspaces-related indicators" such as accessibility index, number of greenspaces, etc. Consider providing a rationale or literature support for choosing these specific indicators.

Figures and Visuals:

 The content mentions figures (e.g., Figure 2, Figure 3) but they aren't provided in the excerpts. Ensure that all figures are clearly labeled, have descriptive captions, and are referenced in the text.

Language & Style:

Proofread the document to ensure there are no repeated sentences or phrases. For instance, the reference "Lutzenhiser, M.; Netusil, N. The Effect of Open Spaces on a Home’s Sale Price. Contemporary Economic Policy 2001, 19, 291–298." appears twice in the references.

Ensure consistent use of terms. For example, if you use "greenspaces" in one section, avoid switching to "green spaces" in another unless there's a specific reason for the distinction.

Reviewer 4 Report

The article "Valuing the Accessibility to Greenspaces in Housing Market: A Spatial Hedonic Analysis in Shanghai, China" discusses a crucial urban studies topic. The methodology is commendably innovative and meticulously implemented. However, there are some notable shortcomings that warrant attention. One concern is the excessive reductionism evident in the study. For instance, the equation of publicly owned and privately owned green spaces contradicts the statement that introduces them, diferentially, leading to critical misinterpretations. Additionally, the use of generic distances to identify catchment areas disregards the distinctive diverse walkability patterns of urban Shanghai, which impacting the overall validity of the findings. The research lacks sufficient validation, and certain elements suffer from basic circular reasoning and truism, raising doubts about the conclusions drawn. Addressing these analytic flaws and ensuring robust validation could strengthen the overall impact and credibility of the study.

Good with some logic slippages.

Reviewer 5 Report

By establishing a spatial hedonic model, this paper comprehensively examines the relationship between residential property values and the accessibility of both community-owned and public greenspaces, which provides guidance for the assessment of residential environmental comfort and has certain academic significance. However, the writing throws the following problems, which can be further improved:

1, The writing of this paper is slightly long and repetitive, and the terms can be condensed and adjusted appropriately;

2, There are some problems with the citation of references, such as " Studies show that golf courses have a positive effect on golf courses [27]" whether the citation of the literature is correctly described, and other please refer to the careful proofreading;

3, This paper shows that for green space larger than 100 square meters to build accessibility index, 30m precision satellite images, it is not clear that how to identify and extract for green space below 1000 square meters?

4, It seems that the article only present public green space for accessibility analysis, for the aforementioned community-owned green space, is it superimposed on the analysis of public green space, or through the community green space rate alone to directly respond to the price of the property?

Moderate editing of English language required.

Round 2

Reviewer 1 Report

Thanks for revising the manuscript accordingly. The discussion can be improved if the authors compare the findings with other relevant studies.

Thanks for revising the manuscript accordingly. The discussion can be improved if the authors compare the findings with other relevant studies.

Reviewer 2 Report

Thanks for your response regarding the use of OLS for the estimation of SAR. I read Lee (2002) and confirmed that OLS is indeed applicable to SAR if the W matrix is given by a dense matrix as assumed in this manuscript. But, Lee (2002) assumed SAR without autoregressive disturbance. Because of that, I still believe that OLS is not available to the author's SAR assuming autoregressive disturbance.

At least, it is needed to reffer Lee (2002) and explain why the authors used OLS. This is important because use of OLS is not common in spatial econometrics.

Line 103: I suggest to replace "GMF-based" with "geographic field model-based" for clearity

Reviewer 4 Report

The second review of the manuscript "Valuing the Accessibility to Greenspaces in Housing Market: A Spatial Hedonic Analysis in Shanghai, China" has identified the persistence of some fundamental flaws in the research. The authors have not addressed any of the concerns raised in the previous review. This is a significant issue, as it suggests that the authors are not taking the feedback seriously and are not committed to improving the quality of their research.

This review also confirmed the authors confuse effects with causes. The authors have simply demonstrated a correlation between green space availability and property value, without considering the reasons why green space is the cause of higher property values: the policy and market driven unequal developmental logics moderating rampant commodification of housing.

Moreover, the revised version has not resolved some serious logic flaws in the manuscript (e.g., the presence of truistic sentences, such as “Studies show that golf courses have a positive effect on golf courses [27].”) that the authors need to address if they want to publish their research in a reputable journal.

I would also suggest that the authors find a more appropriate publisher for their research, such as a journal that specializes in computation methodology for spatial analysis. I am not sure that it is the best fit for your research. A journal that specializes in computation methodology for spatial analysis would be more likely to appreciate the technical aspects of your research and to give it the attention that it deserves.

Required editing to fix erroneous statements, such as "Studies show that golf courses have a positive effect on golf courses." (row 122)
